# Gingival Overgrowths Revealing *PTEN* Hamartoma Tumor Syndrome: Report of Novel *PTEN* Pathogenic Variants

**DOI:** 10.3390/biomedicines11010081

**Published:** 2022-12-29

**Authors:** Samuele Sutera, Daniela Francesca Giachino, Alessandra Pelle, Roberta Zuntini, Monica Pentenero

**Affiliations:** 1Oral Medicine and Oral Oncology Unit, Department of Oncology, University of Turin, San Luigi Gonzaga Hospital, 10043 Orbassano, Italy; 2Medical Genetics, Department of Clinical and Biological Sciences, University of Turin, San Luigi Gonzaga Hospital, 10043 Orbassano, Italy; 3Medical Genetics Unit, Department of Medical and Surgical Sciences, University of Bologna, Policlinico S.Orsola-Malpighi Hospital of Bologna and Center for Studies on Hereditary Cancer, 40138 Bologna, Italy

**Keywords:** gingival overgrowth, *PTEN* hamartoma tumor syndrome, PHTS, Cowden Syndrome, *PTEN*, *PTEN* gene, novel pathogenic variant, hamartoma syndrome, multiple

## Abstract

PTEN hamartoma tumor syndrome (PHTS), is a spectrum of disorders caused by mutations of PTEN, in which non-cancerous growths, called hamartomas, develop in different areas of the body, often including the oral mucosa. PHTS also implies a recognized increased risk of malignancies, as PTEN is a tumor suppressor gene capable of inhibiting progression of several cancers. One of the main and most common clinical manifestation of PHTS are gingival overgrowths presenting as warty lumps. The current study describes patients with gingival or mucosal enlargements leading to the diagnosis of PHTS associated to novel PTEN pathogenic variants. Patients referred to us for gingival lumps suggestive of PHTS associated overgrowths were submitted to genetic analysis in the PTEN gene. Two related and two unrelated patients were investigated. PTEN novel pathogenic variant was found in all of them. Two patients also fulfilled diagnostic criteria of Cowden syndrome (CS). Mucocutaneous lesions, and particularly diffuse gingival overgrowths, are both early and major clinical signs revealing a potential diagnosis of PHTS. Further genetic and clinical assessments are needed in order to confirm and clarify the diagnosis within the PHTS spectrum, including, among others, the CS. A correct interpretation of oral clinical features potentially associated to PHTS is mandatory for diagnosis and a surgical approach can be useful just in case of impairment of periodontal health or for aesthetic needs. The increased risk of malignancies associated to PHTS makes a correct diagnosis pivotal to set up an appropriate lifelong surveillance, aiming at secondary cancer prevention.

## 1. Introduction

In 1999, the term *PTEN* hamartoma tumor syndrome (PHTS) was coined to define a group of disorders characterized by germline *PTEN* mutation and peculiar clinical manifestations (hamartomas overgrowth and increased risk of neoplasia) [1]. Several disorders are part of the PHTS spectrum: Cowden Syndrome (CS), Bannayan–Riley–Ruvalcaba syndrome (BRRS), *PTEN*-related Proteus syndrome (PS) and Proteus-like syndrome, CS being the most frequently observed [2].

Specifically, in the oral cavity PHTS shows up with oral mucosal papillomatosis primarily affecting gingiva and tongue while labial and palatal mucosa are rarely involved. In the literature, the CS/PHTS lesions are described as papules, whitish or pink, small (<4–5 mm), with smooth surfaces, and multiple, that can merge acquiring a typical cobblestone appearance and often associated with a fissured tongue. The pictures below well illustrates such items reported in the literature. Oral manifestations are often the first and earlier signs of disease and lead to the diagnosis, that basically comes from clinical findings [3].

Although oral lesions have, per se, no significant morbidity, PHTS implies an increased risk for malignancies (such as breast, thyroid, and endometrium) [4,5]. Therefore, the early diagnosis allowed by the oral manifestations is important to set an appropriate lifelong follow-up [6].

CS is a rare autosomal dominant inherited condition (prevalence 1:200,000) with multiorgan involvement [7]. It is characterized by a combination of ectodermal, endodermal, and mesodermal changes affecting skin, oral mucosa, gastrointestinal tract, bones, central nervous system, eyes and genitourinary tract [8], so multidisciplinary management should be favored, as recommended in the 2020 Danish guidelines [9].

Over the last two decades a variety of novel pathogenic variants have been discovered, and the number of the reported germline mutations in the *PTEN* gene (at present exceeding 300) has almost tripled [6]. It has been estimated that up to 47% of CS cases are related to novel variants of the *PTEN* gene [10,11].

Current literature on oral manifestations of PHTS usually describe small case series or case reports, more often including patients with CS. Moreover, the differential diagnosis with other conditions, including genetic syndromes, that can express themselves with gingival overgrowth is often discussed [12,13,14,15,16].

We present a series of patients presenting gingival overgrowths leading to the detection of novel loss-of-function *PTEN* pathogenic variant and, in some of these cases, to a final diagnosis of CS/PHTS.

The current paper aimed to report on novel PTEN pathogenic variants and to underline the importance of dental practitioner awareness about oral manifestations of genetic syndromes. Proper knowledge of testing and diagnostic criteria is mandatory in order to obtain early diagnosis and cancer surveillance accordingly. Finally, clinicians should be aware that novel mutations are being uncovered within an evolving frame.

## 2. Materials and Methods

### 2.1. Patients and Clinical Evaluation

Four patients (two related and two unrelated) were referred to the Oral Medicine and Oral Oncology Unit at the San Luigi Hospital, Orbassano, Italy, by dental practitioners for the evaluation of uncommon gingival overgrowths. Physical examination was performed and primarily common conditions leading to generalized gingival enlargement as plaque or drug-induced gingival overgrowth were ruled out. The presence of gingival small multiple whitish or pink papules or nodules or overgrowths, isolated or coalescing in a characteristic cobblestone pattern, in the current NCCN diagnostic/testing criteria for diseases within the PHTS spectrum, were considered when investigating personal and family histories and when genetic testing was performed [3]. The diagnostic/testing criteria are reported below.

### 2.2. Genetic Analysis

The genetic analysis was performed (if not otherwise specified), at the Medical Genetics Unit of the San Luigi Hospital, Orbassano, Italy. Molecular assessment of *PTEN* was performed on DNA extracted from blood samples collected after informed consent. For patient 1, 2, and 3 amplification and direct sequencing were performed for all exons, including the promoter region, as described by Zhou et al. (RefSeq NM_000314.4) [17]. An additional amplification, followed by agarose gel electrophoresis, was conducted to exclude the allelic dropout of the exon 2 due to the polymorphic deletion c.80-956_80-58del occurring in the intron 1, described by Sandel et al. [18]. Oligonucleotide sequences and amplification conditions are available on request. The genomic samples of patients 1 and 2 were sent to the Medical Genetic Unit at S.Orsola-Malpighi Polyclinic Hospital in Bologna (Italy) and analyzed by MLPA (Multiplex Ligation-dependent Probe Amplification, probemix P225 MRC-Holland), searching for duplication or deletion in the *PTEN* gene.

Patient 4 was previously analyzed by a combined SSCP (Single Strand Conformation Polymorphism) and Sanger sequencing approach, at an external diagnostic laboratory.

All the analyses were conducted in a clinical setting, repeating the analyses in an independent sample as part of our laboratory’s usual procedure.

## 3. Results

The patients’ medical histories are summarized in Table 1. Below, each patient is extensively discussed.

### 3.1. Clinical Histories and Management

#### 3.1.1. Patients 1 and 2, Related Patients, Son, and Mother

A 25-year-old man (Patient 1) had been referred to our clinic to rule out a potential diagnosis of Acanthosis Nigricans.

The past medical history was unremarkable, just reporting folliculitis and the patient denied any oncological disorders. The main reason for the medical consultation was the onset of warty lesions involving multiple oral mucosal subsites and the vermilion.

Clinical examination Patient 1

At oral examination, small multiple whitish and pink oral lumps involving gingiva, vermilion, hard palate, and buccal mucosa were observed. The tongue had a fissured aspect with, on the lateral borders, papillomatous lesions in a cobblestone pattern. The lesions were asymptomatic. In addition, the patient had poor oral hygiene and multiple decays (Figure 1).

Even in the presence of poor oral hygiene, lacking massive signs of gingivitis and lacking drug assumption, the oral lesions were evocative of PHTS. Therefore, extra-oral diagnostic criteria for PHTS were investigated. The head circumference was consistent with macrocephaly (61.5 cm; >99th percentile; +4.46 SD). In the presence of at least 2 major criteria, including macrocephaly, the CS/PHTS genetic testing criteria were met, even if clinical diagnostic criteria for CS/PHTS were not [3].

The patient had come to consultation accompanied by his mother. During conversation, lesions involving the mother’s tongue were occasionally detected and she was offered a visit. A potential link between mother and son for inherited disorders had never before been suspected.

Clinical examination Patient 2

At oral examination multiple oral lumps involved the alveolar ridge, buccal mucosa and tongue and oral melanosis was additionally observed. The lesions had a smooth surface, white-pink color and showed a cobblestone pattern. She also had poor oral hygiene.

Extraoral examination showed multiple skin keratotic pits and acral hyperkeratotic papules. The assessment of the head circumference was consistent with macrocephaly (61 cm; >99th percentile; +6.04 SD) (Figure 2).

The 55-year-old woman (Patient 2, mother of Patient 1) outlined her mouth lesions as being long lasting, not responsible for any symptoms and concluded that she had never cared about them.

The remote medical history of Patient 2 revealed appendectomy, uterus malignancy, cerebral ischemia, benign thyroid nodules (leading to partial thyroidectomy), hepatic angioma and four lipomas. Her current medical history was unremarkable.

The family history revealed that Patient 2′s sister had sudden cardiac death and two other siblings were healthy. Their mother was deceased for a not clarified central nervous system malignancy. Their father (smoker) died from larynx cancer and a sister of his had a cervical tumor, another had breast cancer, and a niece too. No other tumors were referred to on the maternal or paternal sides, nor in the older generations.

Patient 2 met the current diagnostic criteria for CS/PHTS. Even when not considering the uterus malignancy, the precise nature of which was undefined, she had two major criteria: macrocephaly and multiple mucocutaneous lesions (oral lumps ≥ 3) and three minor criteria: more than two lipomas, thyroid structural lesions, and vascular anomalies.

Based on this unexpected diagnosis, also Patient 1 fitted the diagnostic criteria for CS/PHTS as a lower number of diagnostic criteria are required for operational diagnosis in a family where at least one individual meets CS/PHTS diagnostic criteria (Table 2, Table 3 and Table 4). In the light of the confirmed CS/PHTS diagnosis for Patient 2, the two major criteria observed in Patient 1 (macrocephaly and oral lumps) led to CS/PHTS diagnosis.

Genetic analyses

The sequencing analysis did not identify any pathogenic mutation in the *PTEN* gene in Patient 1. An intronic heterozygous substitution in intron 6, c.635-245G>C was detected but no evidence supported its role in the splicing process. Nevertheless, this variant was not reported in the GnomAD database [19] and the segregation analysis confirmed that it was inherited from the mother (Patient 2).

The MLPA analysis found the heterozygous deletion of the whole exon 6 c.(492+1_493-1)_(634+1_635-1)del, p.(Gly165Ilefs*9), similar to that reported by Nizialek EA and referred to in the LOVD database [20]. The same deletion was identified in both patients, pointing to a hereditary mutation (Figure 3).

None of the family members were available to extend the genetic examination.

Based on the clinical and genetic results, both patients entered a cancer surveillance program as indicated for PTEN mutation patients (Table 5). The surveillance program includes an annual oral cavity follow-up. So far, the gingival overgrowths had not caused discomfort to the patients and did not require any surgical treatment. Conversely, they entered a professional hygiene program.

#### 3.1.2. Patient 3

A 50-year-old woman (Patient 3) was referred to our clinic by the dentist for the evaluation of multiple gingival warty lesions. The differential diagnosis included squamous papillomas and CS/PHTS manifestations.

Clinical examination

At oral examination, multiple warty lumps were observed involving mainly the gingiva. The most affected area was the lower gingiva at the anterior portion. Of note, the attached gingiva of the central incisors had the typical cobblestone appearance. While the other mucous were free from lesions at the time of the first visit (Figure 4).

At the general medical history, macrocephaly (head circumference 59 cm; >99th percentile; +4.2 SD) and intellectual disability (IQ ≤ 65) were observed, and the patient reported previous surgical asportation of multiple cutaneous acral hyperkeratotic papules, and thyroidectomy for multinodular goiter. Of note, although the mother of the patient (dead at the time of our visit) never received a CS/PHTS diagnosis, the patient was able to report that she had multiple lipomas. In the presence of multifocal oral papillomatosis, associated with macrocephaly (i.e., two or more major criteria; one must be macrocephaly) genetic testing criteria for CS/PHTS were met [3].

Genetic analyses

The sequencing analysis identified a heterozygous deletion in exon 6, c.562del, p.(Tyr188fs).

The pathogenic variant had never been reported before, either in healthy individuals or in CS/PHTS-affected individuals. It causes the production of a truncated protein with a premature stop after 11 codons (p.Tyr188fs). The new protein was likely non-functional, and, according to the in-silico programs, it had a pathogenic effect. The mutation c.562del, p.(Tyr188fs), in conclusion, should be considered a novel pathogenic variant.

Based on this result, the patient entered a cancer surveillance program as indicated for patients with *PTEN* pathogenic variants (Table 5).

Gingival overgrowths never interfered with the patient’s daily habits, and she never had surgical treatments to remove them. The patient was provided with clinical monitoring for the gingival lumps as well as being provided with cancer surveillance.

#### 3.1.3. Patient 4

A 46-year-old man (Patient 4) was referred to our clinic by the dentist for the evaluation and treatment of lumps of the right lateral margin of the tongue and of the mucosal side of the lower lip.

Clinical examination

At the oral examination, in addition to the lesions reported by the dental practitioner, a verrucous aspect and mild hypertrophy of the gingiva surface was also observed. The clinical aspect was of multiple and sessile papules, with smooth surfaces, white–pink color, and of different sizes (<5 mm), showing a cobblestone pattern. 

The patient reported previous excisions with pathological reports consistent with papillomatosis.

The gingival hypertrophy was not consistent with plaque-related disease, although the oral hygiene was poor. Similarly, the drug history was negative, and the gingival hypertrophy could not be drug-related (Figure 5a–c).

The patient presented macrocephaly (head circumference 60.5 cm; >99th percentile; +3.8 SD), cutaneous lesions (crusted nevus at the back, schwannoma (1 cm) on the second toe of the left foot). In his medical history he had thyroidectomy for goiter, and several gastric and colon hyperplastic polyps.

The patient did not meet the revised clinical diagnostic criteria for CS/PHTS, but he already had been submitted to genetic analysis revealing a *PTEN* mutation.

Genetic analyses

Through the Single Strand Conformation Polymorphism (SSCP) and direct sequencing on peripheral blood of the patient, the c.179del mutation, with stop formation at codon 98, was identified p.(Lys60fs). The mutation should be considered pathogenic for PHTS, and tailored cancer surveillance had already been set.

During our clinical follow-up, 4 years after the first assessment, the patient complained about very large lumps of the gingiva, hindering proper oral hygiene, and quite a large overgrowth of the lateral margin of the tongue (Figure 5d–i). We performed surgical excisions, with the restoration of a physiological and hygienically maintainable gingival profile (Figure 5j–o). Due to previous genetic testing revealing the *PTEN* mutation at time of our first visit, the patient was already on a cancer surveillance program when he first came to our clinic.

## 4. Discussion

### 4.1. Diagnostic and Testing Criteria

The *PTEN* gene is located on chromosome 10, locus 10q23.31 and encodes for the protein Phosphatase and Tensin Homolog. It has a crucial tumor suppressor function. The *PTEN* germline mutation has been related to the *PTEN* hamartoma tumor syndrome (Cowden Syndrome and Bannayan–Riley–Ruvalcaba syndrome) and the *PTEN* somatic mutations are frequently found in several tumors. In the 1997 germline *PTEN* mutations were reported in association with CS [21]. Initially, because of small cohorts of patients, *PTEN* mutations were estimated to be present in about 80% of patients, conversely more recent data shows that only 30–35% of patients meeting the CS diagnostic criteria also have a detectable *PTEN* mutation [22,23]. Novel *PTEN* pathogenic variants or other genes mutations could be identified and linked to PHTS or CS in the future, without contrasting with the current diagnostic criteria.

CS and PHTS are rare autosomal dominant inherited conditions. Both definitive diagnoses, as recommendations for genetic testing looking for the PTEN mutation, are based on clinical criteria, implying a pivotal role for physical examination and medical history assessment (Table 2, Table 3 and Table 4) [3,22]. In this scenario, dental practitioners and periodontists may be responsible for diagnosis as gingival overgrowths are one of the most common and earliest clinical manifestations, and they are easily detectable at physical examination.

Dental practitioners should be aware of the PHTS spectrum. Gingival overgrowths not meeting other more common diagnoses should lead to investigation of extraoral features and, eventually, to patient referral for the assessment of the *PTEN* mutation, according to current guidelines.

CS diagnostic criteria were firstly proposed by Salem and Steck in 1983 [24]. In 1996, a consensus of an international consortium of researchers revised them (before the identification of the correlation between *PTEN* and CS) [25].

In 2000 a new version of the International Cowden Consortium operational criteria for the diagnosis of CS was published [26].

In 2013 Pilarski et al., in a systematic review, outlined the available literature including the 2000 Cowden consortium criteria and more recent clinical features. The paper aimed to provide accurate clinical criteria, supported by scientific evidence, to make diagnosis of CS and the spectrum of *PTEN*-related disorders [5].

Current diagnostic criteria belong to the PHTS spectrum where differential diagnosis is also based on the frequency of the signs and on the time of their onset [27]. Thus, CS patients are usually investigated in adulthood, and familial relations can be very useful for an early diagnosis. On the other hand, BRRS occurs in the prenatal era (or immediately after) showing macrocephaly, Hashimoto’s thyroiditis, lipomatosis, vascular malformations, and punctate freckles of the penis or vulva. PS is really variable and shows up with a “mosaic distribution” (involving only some organs or tissues). The clinical diagnostic criteria include mosaic distribution of lesions, sporadic occurrence, a progressive course and additional specific criteria. As a consequence of the variability of the disease manifestation, PS is frequently misdiagnosed [2].

Current diagnostic and testing criteria for CS/PHTS, as reported in Table 2, Table 3 and Table 4, were adopted by the U.S. National Comprehensive Cancer Network (NCCN), in 2017 [22,28], and they are regularly updated and published by the NCCN on a continuous basis, as new clinical and genetic information becomes available. The last revision is the “version 1.2023” [3,29].

Genetic testing criteria are of utmost importance for an early diagnosis and to set up cancer surveillance. In fact, regardless of clinical diagnosis, individuals with a germline *PTEN* pathogenic variant are thought to have the same cancer risks as individuals with CS/PHTS [2]. Major diagnostic criteria of CS/PHTS (according to NCCN guidelines) include several cancers associated to *PTEN* mutation. For this reason, patients who had already experienced malignancy could more easily meet the CS/PHTS diagnostic criteria. Conversely, less strict testing criteria favor an early diagnosis of *PTEN* mutation preceding malignancy and allow the patient to enter specific cancer surveillance (Table 5). Even if a not negligible rate of patients with *PTEN* mutations do not meet diagnostic criteria for CS/PHTS, *PTEN* is known to be a tumor suppressor gene and the patients that show its mutation should be monitored because of the increased risk of developing malignancies [13,28].

Other than the NCCN guidelines, scoring systems have been developed in order to estimate the probability of a *PTEN* mutation. Gene testing is recommended for patients with an estimated probability of 3% or more [30]. According to the “Cleveland Clinic *PTEN* Risk Calculator”, gene testing was recommended for all our patients.

The “Cleveland Clinic *PTEN* Risk Calculator” is a free, user-friendly online calculator able to support clinicians in the assessment of the need for *PTEN* genetic testing. It requires only an accurate anamnesis. When compared to the NCCN guidelines, it has been reported to have a higher sensitivity, thus potentially leading to a higher number of *PTEN* genetic analyses.

The classic NCCN diagnostic criteria have proved effective in adult patients, but it is intuitive that some of the signs might not yet be fully developed in pediatric age groups. Specific diagnostic criteria have been developed for pediatric patients, with the aim of getting an early diagnosis of PHTS (Table 6) [3,31].

The main reasons young patients are brought for medical attention are macrocephaly, neurodevelopmental abnormalities (autism and developmental delay), and dermatologic or mucosal signs (lipomas, oral papillomas) [30,31].

### 4.2. Novel PTEN Variants and Related Gene Function

Three mutations were observed in the PTEN gene from the analysis of our patients, and all mutations were expected to result in a loss of function, due to the premature stop codon.

Patient 1 and 2: deletion of the whole exon 6 c.(492+1_493-1)_(634+1_635-1)del, p.(Gly165Ilefs*9). The deletion of exon 6 was expected to determine a new splicing, for example, between exon 5 and exon 7, with a new reading frame and recognition of a premature stop codon within exon 7.

Patient 3 single nucleotide deletion in exon 6, c.562del, which produced a new reading frame p.(Tyr188fs) and recognized a premature stop after 11 codons. 

Patient 4: the single nucleotide deletion in exon c.179del p.(Lys60fs) induced a new stop codon at codon 98, in exon 5, after a long, mistranslated sequence.

An adjunctive variant c.635-245G>C was identified in patient 1 and his mother, which was a point substitution in the intron between exon 6 and 7, occurring in cis with the pathogenic variant, downstream of it. A direct effect related to such a variant was not expected; it was reported due to its rarity. Accordingly, the nomenclature of the causative deletion could have been changed to c.(492+1_493-1)_(634+1_635-245)del, but it was decided to keep it as is to make the data comparable with the literature of mutations of the PTEN gene.

### 4.3. Clinical Diagnosis

Mouth lesions are described as progressive oral lumps with papillomatous growth, also described as papules, measuring around 1–3 mm, with smooth whitish surfaces, usually on gingiva and tongue and, less frequently, involving labial and palatal mucosa. These lesions often coalesce into confluents sheets, which are described as having a cobblestone appearance [32,33]. Such clinical aspects are usually easily distinguished from inflammatory gingival enlargement, which is characterized by the presence of localized or generalized soft, red, and easy bleeding gingiva in the presence of plaque. Marginal gingiva is invariably involved, and a complete response is obtained after plaque removal. After excluding plaque-induced periodontal diseases, different conditions potentially responsible for the presence of diffuse gingival overgrowth have to be ruled out. In the presence of non-plaque-induced gingival overgrowth a correct diagnosis represents the biggest issue, particularly when facing genetically determined lesions [14]. Several drugs, including immunosuppressants, antihypertensives, anticonvulsant and oral contraceptives, may be responsible for diffuse gingival overgrowth, or it can be caused by hormonal alterations, as occurs in pubertal gingivitis, during pregnancy or secondary to acromegaly. When excluding such scenarios, the presence of an apparently idiopathic fibrous hyperplasia requires a differential diagnosis including genetically determined syndromes, such as tuberous sclerosis, juvenile polyposis syndrome, Peutz–Jeghers syndrome, Birt–Hogg-Dubé syndrome, Gorlin syndrome and neurofibromatosis type 1 [33].

The presence of extra-oral mucosal or cutaneous overgrowths enforces the hypothesis of a genetically determined syndrome. In CS/PHTS clinical manifestations include cutaneous benign hamartoma overgrowth, such as trichilemmomas (yellowish-pink colored or slightly pigmented flat-topped lichenoid or elongated verrucoid cutaneous papules, usually in the face and neck region, ranging from 1 to 5 mm in diameter), acral keratoses (flesh-colored or slightly pigmented smooth or warty papules on the dorsal hands and feet), palmoplantar keratoses (thickening of the skin and scaly spots on the palms and soles) and neuromas. Mucocutaneous lesions are present in 90–100% of patients and often are the first evidence of disease leading to CS/PHTS diagnosis. Commonly, macrocephaly is observed (84%), due to an abnormally enlarged brain (megalencephaly). Such signs are easily detectable at physical examination, but not always enough to meet the CS/PHTS diagnostic criteria [12]. 

All the reported cases had gingival overgrowths not associated to clinical signs suggestive for tissue enlargements induced by plaque-related gingivitis. Even lacking optimal oral hygiene conditions, no gingival redness or tenderness were observed and most of all the tissue overgrowth did not involve just marginal gingiva. Moreover, other oral mucosal subsites were involved, suggesting a syndromic spectrum.

When ruling out a genetically determined condition, an overall medical assessment is needed, including the familiar history. In Patient 1 oral lesions first led to a presumptive diagnosis needing further validation which came after the joint assessment of his mother. This highlights the importance of an overall assessment, including relatives, where rare genetic syndromes are suspected. Such inclusive assessment allows an early diagnosis, which is of paramount importance when considering the increased risk of developing malignancies.

Finally, clinicians should keep in mind that CS/PHTS may be associated with intellectual disability, with potential negative implications limiting the data collection during the diagnostic work-up and compliance in the follow-up. In the reported cases, Patient 2 never minded her massive lumps before our detection and Patient 1, even after the diagnosis, appeared more concerned for esthetic implications at the vermillion than for the increased risk of malignancy.

### 4.4. Oral Management

Periodontists, and, in general, dental practitioners, may have a pivotal role in the early diagnosis of CS/PHTS and, secondarily, they are involved in the local management of gingival overgrowths when needed.

Mucosal lesions, usually leading the patients to seek a consultation, do not have intrinsic morbidity. As observed in the present cases, the oral lumps may have esthetic implications, when involving the vermilion, while gingival overgrowth may impair oral hygiene and sometimes feeding. In such cases, lumps may have to be surgically excised. Similar considerations can apply to skin lesions [4]. 

In detail, oral management includes the following: professional hygiene, to eliminate impairment due to gingival overgrowth; clinical follow-up with an oral medicine practitioner, approximately once a year; and surgical excision of the gingival or mucosal overgrowth performed by a periodontist or an oral surgeon when needed, as previously discussed.

### 4.5. Risk of Malignancies and Surveillance

*PTEN* has been documented to be able to inhibit the progression of several cancers. *PTEN* gene encodes the protein Phosphatase and Tensin Homolog (Pten), which is involved in regulation of the cell cycle thanks to inactivating action of some protein tyrosine kinases. Given that a large number of protein tyrosine kinases work as oncogenes, *PTEN* acts as a major tumor suppressor gene. The downregulation of *PTEN* leads to the known risk of malignancies observed in CS/PHTS [34,35].

The main implication of CS and PHTS is the increased risk of malignancies, to the point that the development of malignancies is part of the diagnostic criteria. Such risk imposes lifelong strict surveillance for secondary oncological prevention [3,36,37,38].

The increased risk of transformation has been reported for several type of cancer: breast cancer (25–85%), thyroid cancer (14–35%), renal cell cancer (17–34%), endometrial cancer (19–29%), colorectal cancer (9–16%) and melanoma (6%) [9]. In light of these numbers, adequate multidisciplinary and tailored surveillance programs for CS/PHTS patients are necessary and specific guidelines can be found, such as the NCCN Guidelines for Cowden Syndrome/*PTEN* Hamartoma Tumor Syndrome Management [3,9,39].

Cancer surveillance should be adopted, not just for CS/PHTS patients (with or without *PTEN* mutation), but also for patients with demonstrated *PTEN* mutation but not fulfilling CS diagnostic criteria.

## 5. Conclusions

The main burden of CS/PHTS is the increased risk of malignancy, conceivably due to *PTEN* inactivation/mutation, so that an early diagnosis is of paramount importance in order to enter tailored surveillance programs. Nevertheless, the diagnosis is currently based only on clinical criteria, including a history of cancer. This could lead to underdiagnosis for several patients showing suggestive clinical signs, but not fitting the current criteria, even in the presence of *PTEN* mutation. Anyway, even lacking a definitive diagnosis, individuals with a germline *PTEN* pathogenic variant are thought to have the same cancer risks as individuals with PHTS and, therefore, should enter a surveillance programs. Other than the NCCN guidelines, the adjunctive use of scoring systems could represent a more sensitive tool in order to recommend genetic testing.

Periodontists, and general dental practitioners, should be aware of gingival overgrowths induced by genetic syndromes, including CS/PHTS, as oral manifestations are one of the most common and early signs which can help in a correct diagnosis and a timely cancer surveillance program. On the other hand, even if the oral mucosa and gingival overgrowths have no intrinsic morbidity, per se, they could significantly affect oral health. Periodontists and dental practitioners should be involved in the multidisciplinary management of the patient, addressing oral lesions and favoring proper oral care and maintenance.

In the future, due to the importance and the oncological implications of PHTS, novel PTEN pathogenic variants should be published and added to the existing database. Furthermore, information about the possible oral manifestations of the genetic syndromes could be spread more, so that dental practitioners can detect them early and refer patients accordingly, thereby improving their prognosis.

## Figures and Tables

**Figure 1 biomedicines-11-00081-f001:**
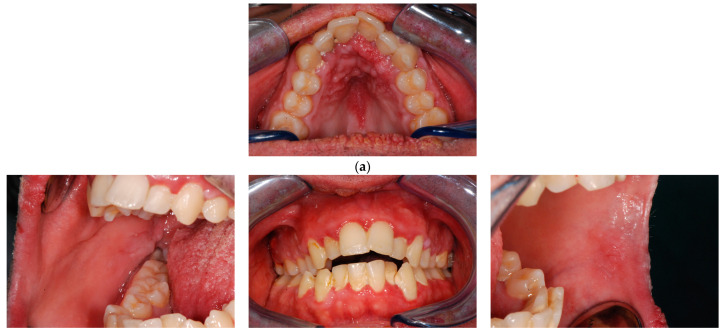
Patient 1, clinical lesions. (**a**) Palate. (**b**) Right buccal mucosa. (**c**) Gingiva. (**d**) Left buccal mucosa. (**e**) Right lateral lingual border. (**f**) Tongue. (**g**) Left lateral lingual border. (**h**) Labial vermilion and perioral skin.

**Figure 2 biomedicines-11-00081-f002:**
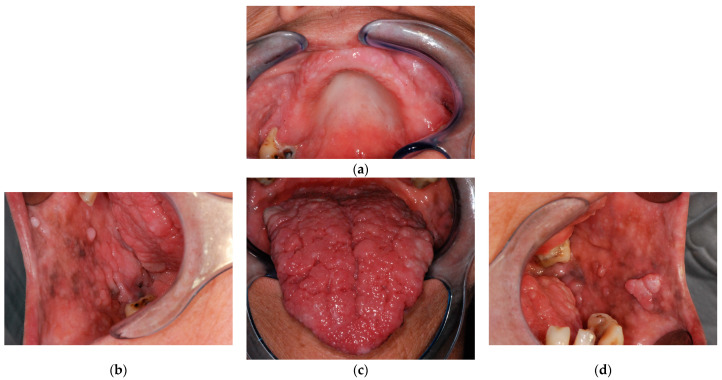
Patient 2, clinical lesions. (**a**) Palate. (**b**) Right buccal mucosa. (**c**) Tongue. (**d**) Left buccal mucosa. (**e**) Right lateral lingual border. (**f**) Ventral tongue. (**g**) Left lateral lingual border. (**h**–**j**) skin lesions, foot. (**k**) Inframammary fold.

**Figure 3 biomedicines-11-00081-f003:**
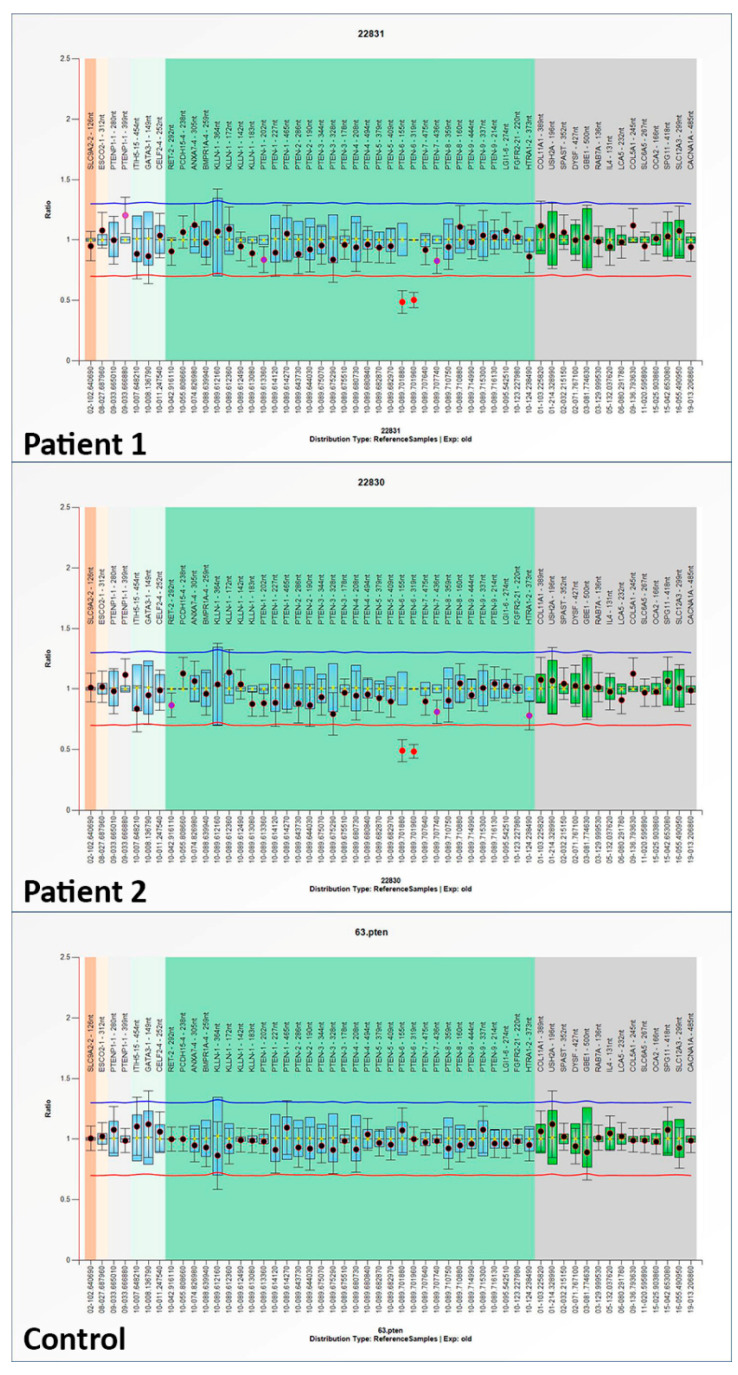
Genetic analysis revealing the same pathogenic variant in Patient 1 and 2. The MLPA analysis was conducted with Coffalyser software. The 95% confidence interval of a probe over the reference samples is depicted as a colored bar in the ratio chart. Both PTEN exon 6 in Patient 1 and in Patient 2 resulted under the arbitrary lower border (red line) which is set at 0.65 (0.95–0.3) and thus they are considered deleted in the samples. The blue line which is set at 1.25 (0.95 + 0.3) represents the upper arbitrary border. If a probe report crossed the latter boundary, this would be indicative of a duplication.

**Figure 4 biomedicines-11-00081-f004:**
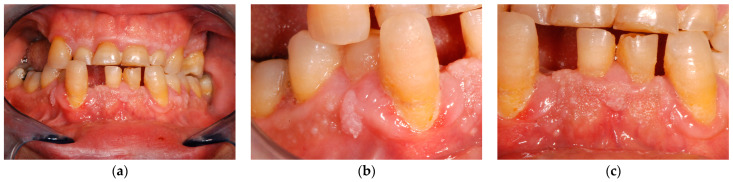
Patient 3, clinical lesions. (**a**) Overall view. (**b**,**c**) Gingival lumps details.

**Figure 5 biomedicines-11-00081-f005:**
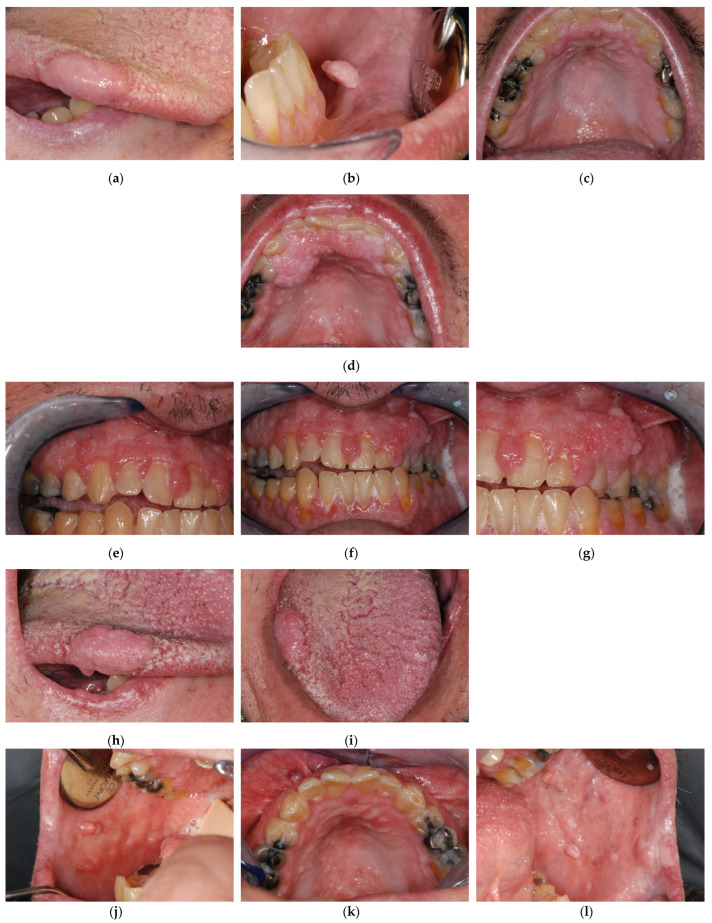
Patient 4 clinical history. First evaluation (**a**) Right lateral lingual border. (**b**) mucosal side of the lower lip. (**c**) Palate. After 4 years follow-up patient complained about very marked gingival hypertrophy (**d**) Palate. (**e**–**g**) Gingival aspect. (**h**,**i**) Tongue lesion. (**j**–**o**) mucogingival plastic surgical approach results.

**Table 1 biomedicines-11-00081-t001:** CS/PHTS clinical diagnostic criteria and genetic analysis in the reported patients.

	Patient 1	Patient 2	Patient 3	Patient 4
Major Criteria				
Multiple mucocutaneous lesions	Yes	Yes	Yes	Yes
Macrocephaly	Yes	Yes	Yes	Yes
Minor criteria				
Lipomas (≥3)	No	Yes	No	No
Intellectual disability	No	No	Yes	No
Thyroid structural lesions	No	Yes	Yes	Yes
Vascular anomalies	No	Yes	No	No
Symptoms	No	No	No	Mechanical oral discomfort
Family history of PHTS/CS	Yes	No	No	No
Genetic Analysis	*PTEN* mutation	*PTEN* mutation	*PTEN* mutation	*PTEN* mutation
Diagnosis	Cowden Syndrome	Cowden Syndrome	*PTEN* mutation	*PTEN* mutation
Cancer Surveillance	YES	YES	YES	YES

**Table 2 biomedicines-11-00081-t002:** *PTEN* hamartoma tumor syndrome clinical diagnostic criteria.

Major Criteria	
Breast cancer	
Endometrial cancer	Epithelial
Thyroid cancer	Follicular
Gastrointestinal hamartomas (≥3)	Including ganglioneuromas, but excluding hyperplastic polyps
Lhermitte-Duclos disease	Adult
Macrocephaly	≥97th percentile: 58 cm for females, 60 cm for males
Macular pigmentation of the glans penis	
Multiple mucocutaneous lesions OR biopsy proven OR dermatologist diagnosed (any of the following)	Multiple trichilemmomas (≥3, at least one biopsy proven) Acral keratoses (≥3 palmoplantar keratotic pits and/or acral hyperkeratotic papules) Mucocutaneous neuromas (≥3) Oral papillomas (particularly on tongue and gingiva), multiple (≥3)
**Minor Criteria**	
Autism spectrum disorder	
Colon cancer	
Esophageal glycogenic acanthosis (≥3)	
Lipomas (≥3)	
Intellectual disability (ie, IQ ≤ 75)	
Renal cell carcinoma	
Testicular lipomatosis	
Thyroid cancer	papillary or follicular variant of papillary
Thyroid structural lesions	e.g., adenoma, multinodular goiter
Vascular anomalies	including multiple intracranial developmental venous anomalies

**Table 3 biomedicines-11-00081-t003:** Testing criteria for Cowden Syndrome/*PTEN* Hamartoma Tumor Syndrome [3].

Testing is Clinically Indicated in the Following Scenarios
Individual from a family with a known *PTEN* P/LP variant Individual with a personal history of BRRS Individual meeting clinical diagnostic criteria for CS/PHTS Individual not meeting clinical diagnostic criteria for CS/PHTS with a personal history of: → Adult Lhermitte-Duclos disease (cerebellar tumors); or → Autism spectrum disorder and macrocephaly; or → Two or more biopsy-proven trichilemmomas; or → Two or more major criteria (one must be macrocephaly); or → Three major criteria, without macrocephaly; or → One major and at least three minor criteria; or → At least four minor criteria At-risk individual with a relative with a clinical diagnosis of CS/PHTS or BRRS for whom testing has not been performed → The at-risk individual must have the following: ▪ Any one major criterion or ▪ Two minor criteria *PTEN* P/LP variant detected by tumor genomic testing on any tumor type in the absence of germline analysis

For general testing criteria not listed above consult NCCN Guidelines Version 1.2023. (https://www.nccn.org/professionals/physician_gls/pdf/genetics_bop.pdf access on 27 December 2022)

**Table 4 biomedicines-11-00081-t004:** Operational diagnosis.

Single Individual	
Three or more major criteria	Including at least one between: macrocephaly, Lhermitte-Duclos disease, gastrointestinal hamartomas
Two major and three minor criteria	
**Family History for the Condition or Has a *PTEN* Mutation**	
Any two major criteria with or without minor criteria	
One major and two minor criteria	
Three minor criteria	

**Table 5 biomedicines-11-00081-t005:** Danish guidelines for the cancer surveillance of patients with germline *PTEN* mutation.

Surveillance	Recommendations
Physical examination	Physical, neurological, and cognitive examination at the time of diagnosis. Consider follow-up depending on patient symptoms and age
Breast	Annual clinical mammography starting at age 30. Consider MRI
Thyroid	Baseline ultrasound and clinical examination at age 15, from the age of 18, annually
Endometrial	Annual transvaginal ultrasound starting at age 30–35. Consider endometrial biopsy
Renal cell	US or MRI of the kidneys every second year starting at age 40
Colon	Colonoscopy every 5 years starting at age 35
Melanoma	Dermatologic evaluation at the time of diagnosis and repeated based on individual assessment

Table with inspiration from Smerdel et al. 2020, European Journal of Medical Genetics.

**Table 6 biomedicines-11-00081-t006:** Pediatric criteria for diagnosis of *PTEN* Hamartomas tumor syndrome.

Required Criterion	Secondary Criteria
Macrocephaly (≥2 standard deviations)	At least one of the following: Autism spectrum disorder or developmental delay Dermatologic features (lipomas, oral papillomas, trichilemmomas, penile freckling) Vascular features (arteriovenous malformations or hemangiomas) Gastrointestinal polyps Pediatric-onset thyroid cancer or germ cell tumors

Table with inspiration from Yehia, Keel, Eng 2020, Annual Review of Medicine.

## Data Availability

The data presented in this study are available on request from the corresponding author. The data are not publicly available due to privacy.

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
