# Peer review of "Gingival Overgrowths Revealing *PTEN* Hamartoma Tumor Syndrome: Report of Novel *PTEN* Pathogenic Variants"

_biomedicines, 2022, doi:10.3390/biomedicines11010081_

Round 1

Reviewer 1 Report

New variants of PTEN are discovered which should be added to the literature and are of great importance.

Minor improvements:

1. use a table to list the patient information and symptoms and main findings so that it is easy for the readers to grasp the main findings

2. discuss how the substitute variants may affect PTEN function -- loss of function for truncated form is obvious, but how about the other forms? Will these substitutes potentially lead to compromised PTEN function?

Author Response

Thank you for your revision, we appreciate your suggestions.

Our responses: 

1: We added the table (new table 1)

2: We discussed the topic in the new paragraph 4.2

I'm also attaching a file with our responses

Reviewer 2 Report

First of all I would like to congratulate the authors regarding the scientific work. The article brings new pieces of the puzzle called oral manifestation of the genetic syndromes. 

The article is very well structured, with important informations regarding PTHS. The sections of the article are written in a clearly way. I appreciate the patient photos that can be used as a guide for the clinicians.

My recommendations are:

1. if you have available a more comprehensive oral examination on the cases, please describe it in the clinical examination section

2. the article must be revised by a native english speaker.

Author Response

Thank you for your revision, we appreciate your suggestions.

Our responses: 

1: We implemented the oral examination section of each patient.

2: Given the short time we had to modify the manuscript we didn't get the article reviewed by a native English speaker yet. If it is required after the evaluation of the minor revision we made, we will submit the proofreading of the manuscript to an English native.

I'm also attaching a file with our responses.

Reviewer 3 Report

Dear Authors,

The manuscript under review attempts to present a case series on Gingival overgrowths revealing PTEN hamartoma tumor syndrome. Reporting of novel PTEN pathogenic variants. The manuscript did capture the details of the cases, including the examination, diagnosis, clinical findings, and treatment. All the manuscript sections are well written and concluded but need a few corrections. Kindly find below the comments and suggestions, which will help the authors check and revise the manuscript.

Abstract:

1.       Kindly write the abstract as a single paragraph

2.       Should be structured but without headings

3.       Kindly delete the discussion section

4.       Provide MesH keywords

Introduction:

1.       Briefly write about the oral manifestation of PTEN hamartoma tumor syndrome (PHTS)

2.       If possible with pictures

3.       Write about the objective of the current case series

4.       Write about literature on the oral manifestation of PHTS

Materials and methods:

1.       Kindly remove the material and methods, and results sections and change to case presentation

2.       Genetic analysis: Did the authors perform the confirmatory test?

3.       Patient 4 was previously analyzed by a combined SSCP (Single Stand Conformation Polymorphism) and Sanger sequencing approach: provide the reference

4.       Who performed the clinical examination and genetic analysis? Authors?

Discussion:

1.       A very well-written discussion

2.       4.3. Oral management: kindly write in detail

3.       Kindly add clinical importance in relation to oral health and future recommendation 

Author Response

Thank you for your revision, we appreciate your suggestions.

Our responses: 

Abstract:

1, 2, 3: Thank you for your suggestion, but we have structured the abstract following the journal template, that require headings (including discussion section).

  1. We have added “Hamartoma Syndrome, multiple”, whose entry terms include among others multiple hamartoma syndrome(s), Cowdens Disease, Cowden Syndrome, PTEN Hamartoma Tumor Syndrome, Macrocephaly, Multiple Lipomas, and Hemangiomata

Introduction:

  1. We have implemented the description in the lines 44 to 49.
  2. Pictures showing the overall clinical examination of the reported cases have been included in the paper. These pictures can well describe oral manifestation as previously described in the literature.
  3. We have added it. Please see the last introduction paragraph
  4. We have written about. Pleae see the lines 65 to 68.

Materials and Methods:

  1. Thank you for your suggestion, but the current structure with material and methods, and results sections was a specific and direct request of the Academic Editor, instead of a “case presentation”.
  2. We have specified in the text. Lines 109-110.
  3. We have specified in the text. Line 108: “at an external diagnostic laboratory”.
  4. Yes, we did. We have specified in “Materials and Methods” section as much as in “Author Contributions”.

Discussion:

  1. Thank you.
  2. “4.3”, that in the new version has become “4.4 Oral management”. We have implemented the section.
  3. We have implemented the “conclusions” section.

I'm also attaching a file with our responses.
